# Land Consumption and Land Take: Enhancing Conceptual Clarity for Evaluating Spatial Governance in the EU Context

Elisabeth Marquard [1,*], Stephan Bartke [1], Judith Gifreu i Font [2], Alois Humer [3], Arend Jonkman [4], Evelin Jürgenson [5], Naja Marot [6], Lien Poelmans [7], Blaž Repe [8], Robert Rybski [9], Christoph Schröter-Schlaack [1], Jaroslava Sobocká [10], Michael Tophøj Sørensen [11], Eliška Vejchodská [12,13], Athena Yiannakou [14] and Jana Bovet [15]

1 Helmholtz Centre for Environmental Research—UFZ, Department of Economics, Permoserstraße 15, 04318 Leipzig, Germany; stephan.bartke@ufz.de (S.B.); christoph.schroeter-schlaack@ufz.de (C.S.-S.)
2 Faculty of Law, Universitat Autònoma de Barcelona, Bellaterra (Cerdanyola del Vallès), 08193 Barcelona, Spain; judith.gifreu@uab.cat
3 Department of Geography and Regional Research, University of Vienna, Universitaetsstrasse 7/5, 1010 Vienna, Austria; alois.humer@univie.ac.at
4 Department of Management in the Built Environment, Delft University of Technology, Julianalaan 134, 2628BL Delft, The Netherlands; a.r.jonkman@tudelft.nl
5 Chair of Geomatics, Institute of Forestry and Rural Engineering, Estonian University of Life Sciences, Kreutzwaldi 5, 51014 Tartu, Estonia; evelin.jyrgenson@emu.ee
6 Biotechnical Faculty, Department of Landscape Architecture, University of Ljubljana, Jamnikarjeva ulica 101, 1000 Ljubljana, Slovenia; naja.marot@bf.uni-lj.si
7 VITO—Vlaamse Instelling voor Technologisch Onderzoek, Unit Ruimtelijke Milieuaspecten, Boeretang 200, 2400 Mol, Belgium; lien.poelmans@vito.be
8 Faculty of Arts, Geography Department, University of Ljubljana, Aškerčeva cesta 2, 1000 Ljubljana, Slovenia; blaz.repe@ff.uni-lj.si
9 Faculty of Law and Administration, University of Warsaw, Krakowskie Przedmieście 26/28, 00927 Warsaw, Poland; robert.rybski@wpia.uw.edu.pl
10 National Agricultural and Food Centre—Soil Science and Conservation Research Institute, Trenčianska 55, 821 09 Bratislava, Slovakia; jaroslava.sobocka@nppc.sk
11 Technical Faculty of IT and Design, Department of Planning, Aalborg University, Rendsburggade 14, 9000 Aalborg, Denmark; tophoej@plan.aau.dk
12 Faculty of Social and Economic Studies, Institute for Economic and Environmental Policy, Department of Economics and Management, Jan Evangelista Purkyně University, Moskevská 54, 400 96 Ústí nad Labem, Czech Republic; eliska.vejchodska@ujep.cz
13 Faculty of Humanities, Department of Social and Cultural Ecology, Charles University, Pátkova 2137/5, 182 00  Prague 8, Czech Republic
14 Faculty of Engineering, School of Spatial Planning and Development, Aristotle University of Thessaloniki, 54124 Thessaloniki, Greece; adgianna@plandevel.auth.gr
15 Helmholtz Centre for Environmental Research—UFZ, Department of Environmental and Planning Law, Permoserstraße 15, 04318 Leipzig, Germany; jana.bovet@ufz.de
* Correspondence: lisa.marquard@ufz.de; Tel.: +49-341-235-1835

**Abstract:** Rapid expansion of settlements and related infrastructures is a global trend that comes with severe environmental, economic, and social costs. Steering urbanization toward well-balanced compactness is thus acknowledged as an important strategic orientation in UN Sustainable Development Goal 11 (SDG-11) via the SDG-indicator "Ratio of land consumption rate to population growth rate." The EU's simultaneous commitment to being "a frontrunner in implementing [ . . . ] the SDGs" and to striving for "no net land take until 2050" calls for relating the concepts of *land consumption* and *land take* to each other. Drawing on an EU-centred questionnaire study, a focus group and a literature review,

we scrutinize definitions of *land consumption* and *land take*, seeking to show how they are interrelated, and questioning the comparability of respective indicators. We argue that conceptual clarifications and a bridging of the two notions are much needed, and that the precision required for definitions and applications is context-dependent. While approximate understandings may suffice for general communication and dissemination objectives, accurate and consistent interpretations of the discussed concepts seem indispensable for monitoring and reporting purposes. We propose ways of addressing existing ambiguities and suggest prioritizing the term *land take* in the EU context. Thereby, we aim to enhance conceptual clarity around land consumption and land take—a precondition for solidly informing respective policies and decisions.

**Keywords:** artificialization; compact cities; densification; indicators; SDG-11; soil sealing; spatial planning; sustainable development; urbanization

---

## 1. Introduction

The conversion of undeveloped land into built-up urban areas, or to industrial sites and transport infrastructure is an underrated, yet increasing problem worldwide [1,2]. Termed *land consumption* or *land take*, it results in the loss of multifunctional, fertile soils and in the deterioration of biodiversity and ecosystem services [3–5]. Moreover, the exceeding growth of the built environment poses further challenges, e.g., related to health, housing, transport, or energy provision and related carbon emissions [6] (pp. 26–28), as well as to public services provision [7]. Steering cities toward well-balanced compactness is, therefore, an important strategy to level their environmental footprint, to support culturally and aesthetically attractive downtown areas, and to secure adequate public infrastructure at viable costs [8–10]. UN Sustainable Development Goal 11 (SDG-11), calling for inclusive, safe, resilient, and sustainable cities, covers the spatial aspect of urbanization by its indicator of land consumption (SDG indicator 11.3.1: "Ratio of land consumption rate to population growth rate"). The indicator postulates that where rates of land conversions due to urban development are disproportionately high compared to population dynamics, such a "growth turns out to violate every premise of sustainability that an urban area could be judged by" [11].

In the European Union (EU), almost three quarters of the population live in cities, towns, and suburbs, and further urbanization is expected [12] (p. 25). The EU promotes land recycling, e.g., via brownfield redevelopment [13], but such re-use of land accounts only for a small fraction of urban development so far. During the period 2000–2018, eleven times more land was newly developed than re-cultivated within the EU [14]. Built-up areas have expanded predominantly at the expense of agricultural land with mostly high-quality soils [1,14] and, during the past decades, expansion has not corresponded with urban population dynamics [15] (p. 11). In fact, between 1990–2015, Europe was the world region with the least efficient use of urban land: while the population increased in these years by 2.4%, the built-up areas expanded by more than 30% [16]. Acknowledging that this trend is unsustainable and that it risks fostering inexpedient urban sprawl with all its negative side-effects, the EU introduced the soil sealing guidelines [13] and declared the goal of "no net land take" within the EU by 2050 [17,18]. This objective thus relates very closely to the SDG-11 indicator of land consumption. Both imply that the conversion of undeveloped into developed land should be minimized. However, besides this obvious overlap with regard to content, there is a striking difference in terminology, prompting the questions of what land consumption and land take exactly entail, whether both terms are synonymous, and how they relate to other important concepts such as soil sealing and artificialization.

A clear conceptualization of the terms land consumption and land take is not only of academic interest but of high practical and political relevance. It is essential for effectively exchanging knowledge on spatial governance nationally and internationally, and also for compatible monitoring and reporting, which in turn impacts the credibility of quantitative policy targets [19,20] (p. 45). A need for clarification

and guidance has been highlighted repeatedly see, e.g., [19,21,22], and with regard to the EU policy context, it has been pointed out how: "to date, there is a lack of harmonised terminologies across [EU] M[ember] S[tates] to describe issues related to the development of artificial areas as well as a lack of consensus regarding how to assess their environmental and socio-economic impacts and how to define compensation" [20] (p. 56 and footnote 23). The following paragraphs outline the two internationally agreed indicators used in this context.

The SDG-11 indicator of land consumption was adopted by the UN General Assembly in 2017 as part of the SDG framework. This framework is used for producing comparable information on how countries perform with respect to fundamental individual and societal human needs and is meant to be complemented by national indicators decided upon autonomously by the UN Member States. Every SDG is underpinned with several SDG targets to which the indicators are assigned. In the case of SDG-11, target 11.3 reads: "By 2030, enhance inclusive and sustainable urbanization and capacity for participatory, integrated and sustainable human settlement planning and management in all countries" and one of its indicators is the "ratio of land consumption rate to population growth rate." If the indicator value is < 1, the increase in built-up area is slower than the increase in the number of inhabitants, and it can be explained by the demographic trend; if, instead, the indicator value is > 1, land consumption is disproportionately high, i.e., decoupled from population growth.

The EU-wide land take indicator was established around 2004 by the European Environment Agency (EEA) [23]. It is regularly monitored and reported by the EEA [24–26], and this has significantly increased the political relevance of the term land take in the EU. Specifically, its inclusion as a key concept in several strategic EU documents e.g., [17,18] anchored it in the political discourse on spatial development in Europe [27]. At the same time, land consumption continues to be used as a key concept in policy- or science-policy documents related to land governance and with relevance to the EU e.g., [6,28–30].

With this study, we aim to further scrutinize the terms land consumption and land take from an EU-centred perspective, seeking to show how they are interrelated. We argue that the formation of a common understanding of these terms and their relationship (also with other concepts), as well as the testing of their practical application, lags behind their increasing use and relevance in policy-settings. With an analytical emphasis on the land take concept, we present 30 published definitions of relevant terms and reflect on major definition-related ambiguities and practical impediments. By providing this overview, we aspire to contribute to the discourse on sustainable land governance, particularly within the EU.

## 2. Materials and Methods

This study draws on an in-depth questionnaire survey, a focus group meeting, and a review of scientific literature and policy-relevant documents. All three research components were performed during 2018–2020 as part of the project "SURFACE—Standards and Strategies for the reduction of land consumption" (https://www.ufz.de/surface/). They were performed to investigate how land consumption and land take is conceived and dealt with at the EU level, as well as on the national level in several EU- and selected non-EU countries. Thus, taken together, the three research components had a wider thematic scope than this article and covered several aspects of the political and legal governance of land conversions, and the political and scientific discourse on the matter. For this article, we selectively processed information related to the conceptualization of key terms and on approaches for assessing land consumption and land take via indicators at the international and national scale. Our research was clearly EU-centred, resulting in a bias toward the EU policy context and the concept of land take.

### 2.1. Recruitment of Participating Experts

For the questionnaire survey and the focus group, we recruited experts by purposive sampling [31] (p. 422) and applied the following selection criteria: pertinent expertise in land use issues, professional

affiliation, and geographic localization. We endeavoured to include experts with scientific, administrative, juridical, and political knowledge, and therefore approached representatives of academic and policy-related institutions. Furthermore, we strived to cover as many EU countries as possible, and, in the case of the questionnaire survey, of selected non-EU countries for comparison. Potential candidates were identified primarily via pre-established contacts and collaborations on the respective topic, and in some cases by screening the authors of relevant documents or the members of respective institutions. For the questionnaire survey, 48 persons were contacted in 30 different countries (of which 22 were EU member states). Of those contacted, 25 experts responded, covering 20 countries. In addition, two of the researchers running the SURFACE project completed the questionnaire for Germany which resulted in a sample of 21 countries (16 EU and five non-EU countries, see Appendix A Table A1). The disciplinary background of the respondents was diverse, the majority holding degrees in law, spatial planning or geography, and their affiliations were either scientific or administrative institutions responsible for or dealing routinely with the matter at hand.

The focus group was formed by 18 experts from 13 EU countries who were joined by three members of the SURFACE team. Twelve of these experts had previously taken part in the questionnaire survey. All but two authors of this article participated in the focus group meeting.

*2.2. Data Collection*

The questionnaire was distributed by email and centred on the issue of land take (this term was used by the research team, see Supplementary Materials Table S1 for details of the questionnaire design). The participants were asked to describe the respective situation in the country they were most knowledgeable about (usually the country of their professional workplace). The conceptualization of land take and related terms was highly significant for many of the questions posed, in particular those that dealt with indicators and monitoring systems. Anticipating the need for clarifying the terminology, the EEA-definition of the term land take was provided to the participants in the introduction of the questionnaire as a reference point [32] (see also definition #10 in Table 1) which ensured that the majority of them interpreted land take in the intended way. However, the answers obtained also showed marked differences in the familiarity with the land take concept across survey participants, e.g., two respondents associated it with land expropriation and one with the purchase of land by foreign investors ("land grabbing"). Additionally, many respondents pointed out that the concept was not the one they normally use, and some provided equivalent or closely related terms in their own languages.

**Table 1.** Definitions of (a) land consumption, (b) land take, and (c) related terms, in chronological order of cited references (if publication dates were apparent); accentuations added.

| # | Definition | Reference |
|---|---|---|
| | **(a) Land Consumption:** | |
| 1 | **Land consumption includes:** (a) The expansion of built-up area which can be directly measured; (b) the absolute extent of land that is subject to exploitation by agriculture, forestry, or other economic activities; and (c) the over-intensive exploitation of land that is used for agriculture and forestry. | [11], similar in EEA Glossary [1] |
| 2 | **Land consumption rate:** the annual rate at which cities uptake land for urbanized uses (both built-up and open space demands) | [33] |
| 3 | [ . . . ], the percentage of current total urban land that was newly developed (consumed) will be used as a measure of the **land consumption rate**. The fully developed area is also sometimes referred to as built up area. | [11] |
| 4 | [ . . . ] the developed area per capita, [which] we refer to as "per capita **land consumption.**" | [34] |
| 5 | The '**land consumption**' (percentage) is measured as the percentage share of urban (artificial) land to the total land area. It indicates the level of urbanisation for a given area. | [35] |
| 6 | The [LUISA [2]] land use intensity indicator measures the **land consumption** or the size of actual artificial areas per inhabitant, expressed in square meters per inhabitant. | [12] (p. 34) |
| 7 | The seven sources of information compared in this paper refer to the **three forms of land "consumption"** evoked earlier: **land take**, soil sealing, and building plots. | [19] |

**Table 1.** *Cont.*

| # | Definition | Reference |
|---|---|---|
| | **(b) Land Take:** | |
| 8 | In a more general sense, **land consumption** can be considered the change from a non-artificial land cover to an artificial land cover of the ground [ . . . ]. | [36] |
| 9 | **Land take:** The area of land that is "taken" by infrastructure itself and other facilities that necessarily go along with the infrastructure, such as filling stations on roads and railway stations. | EEA Glossary [1] |
| 10 | The **land take indicator** addresses the change in the area of agricultural, forest, and other semi-natural and natural land taken for urban and other artificial land development. Land take includes areas sealed by construction and urban infrastructure, as well as urban green areas, and sport and leisure facilities. | [14], similar in [23,32] |
| 11 | **Land take**, also referred to as **land consumption**, describes an increase of settlement areas over time. This process includes the development of scattered settlements in rural areas, the expansion of urban areas around an urban nucleus (including urban sprawl), and the conversion of land within an urban area (densification). | [13], similar in [37,38] [3] |
| 12 | [ . . . ], "**Land take**" is defined as the amount of agriculture, forest, and semi-natural land taken by artificial land [ . . . ]. | [39] (p. 12) |
| 13 | [ . . . ] the "**net land take**" concept could be subject to different interpretations. It can be defined "arithmetically" as "changes of non-artificial areas into artificial areas, which are not compensated by the restoration of the same amount of artificial areas into non-artificial areas" or in a more "ecological" manner depending on the balance between the land functions lost and restored. | [20] (p. 35) |
| 14 | **Gross land take** [is] defined as the growth of artificial areas irrespective of re-naturalisation. | [20] (p. 35) |
| 15 | One measure of urban development is the "**land take**" (i.e., the amount of land converted into artificial or built-up areas) [ . . . ]. | [12] (p. 30) |
| 16 | The concept of **land take** covers all forms of conversion for the purpose of settlement, including: the development of scattered settlements in rural areas; the expansion of urban areas around an urban nucleus; the conversion of land within an urban area (densification); and the expansion of transport infrastructure such as roads, highways, and railways. Broadly, this discussion considers as **land take** any conversion of agricultural, natural, or semi-natural land cover to an "artificial" (e.g., human-made) area. | [38] (p. 65) |
| 17 | **Land take**, by its definition, is the subtraction of an area from a previous agricultural, natural or semi-natural land use. | [38] (p. 67) |
| 18 | **Land take:** Converting agricultural or forestland or natural habitats to residential, industrial, commercial, or infrastructure areas. | [40] |
| 19 | **Land take** is the process in which urban areas and sealed surfaces occupy agricultural, forest, or other semi-natural and natural areas. | [25] |
| 20 | **Average annual land take** (the increase of artificial land) | [41] |
| 21 | Settlement area per capita: This indicator captures the amount of settlement area due to **land take** such as for buildings, industrial and commercial areas, infrastructure, sport grounds, etc., and includes both sealed and non-sealed surfaces. | [42] |
| | **(c) Related Concepts:** | |
| 22 | **Soil sealing** refers to changing the nature of the soil such that it behaves as an impermeable medium (for example, compaction by agricultural machinery). Soil sealing is also used to describe the covering or sealing of the soil surface by impervious materials by, for example, concrete, metal, glass, tarmac, and plastic. | EEA Glossary [1] |
| 23 | **Soil sealing** is the loss of soil resources due to the covering of land for housing, roads, or other construction work. | [43] (p. 110) |
| 24 | The covering of the soil surface with impervious materials as a result of urban development and infrastructure is known as **soil sealing**. | [43] (p. 110) |
| 25 | **Soil sealing** means the permanent covering of an area of land and its soil by impermeable artificial material (e.g., asphalt and concrete), for example through buildings and roads. | [13] |
| 26 | **Sealing of land areas** indicates the amount of area covered with impervious materials due to urban development, increases in traffic infrastructure and construction (for example, buildings, constructions, and laying of completely or partially impermeable artificial material, such as asphalt, metal, glass, plastic, or concrete). | [41] (p. 296) |
| 27 | [ . . . ] **artificial areas** are defined as urban fabric, industrial/commercial land uses and infrastructures/transport networks. | [39] (p. 12) |
| 28 | **Artificial land** is defined as the total of artificial non-built up areas (such as parking lots, playgrounds, farms, cemeteries, roads, railways, and bridges) as well as built-up areas (for example, buildings and greenhouses). | [44] (p. 287) |
| 29 | **Artificialised land** is the variety of land use supporting all human activities other than agriculture and forestry | [21] (p. 151) |
| 30 | **Artificial surfaces** = land that is assigned to one of the following classes: urban fabric (continuous and discontinuous); industrial, commercial, and transport units; mine, dump, and construction sites; artificial, non-agricultural vegetated areas (green urban areas, sport, and leisure facilities). | [45] |

[1] https://www.eea.europa.eu/help/glossary#c4=10&c0=all&b_start=0. [2] LUISA stands for the 'Land-Use-Based Integrated Sustainability Assessment' modelling platform, see https://ec.europa.eu/jrc/en/luisa. [3] This definition was taken up by a workshop of the European Committee for Standardization (CEN Workshop 74), i.e., it became part of an early stage standardisation activity; the draft is available at ftp://ftp.cencenelec.eu/CEN/Sectors/List/Environment/HOMBRE/DraftCWA74_20140528_.pdf.

The challenge of finding a common understanding of land consumption and land take also pervaded the discussions during the focus group meeting. This two-day event was held in April 2019 in Berlin/Germany and provided room for discussing central issues and ideas related to the content of the questionnaire survey. Five guiding questions were provided to the participants prior to the meeting, inviting them to reflect on the particular situation in their country and on their (national) perspective on the following issues in the context of land consumption and land take: problem awareness among different stakeholder groups, the course and actors of relevant political debates, relevance of SDG-11 and SDG-15, monitoring programs and indicator sets, effectiveness of policies, and ideas for improvement.

For the literature review, we considered scientific articles dealing with spatial governance or related themes, as well as documents relevant to the respective EU policy context, and some pertinent internet sources (see Supplementary Materials Table S2 for a list of the 50 policy relevant documents considered, all released during the period 1997–2019). We extracted published definitions, interpretations, and limitations of the concepts land consumption and land take, and, less exhaustively, of the concepts soil sealing and artificialization. This imbalance was due to the fact that land consumption and land take were in the focus of this study, while soil sealing and artificialization were regarded as important additional concepts related to land consumption and land take.

While insights on the conceptualization and monitoring of land take could be gained through all three research components of this study (questionnaire survey, focus group, and literature review), respective information on the related concepts land consumption, soil sealing, and artificialization was mostly retrieved from the literature.

*2.3. Data Analysis*

To explore how the concepts of interest have been defined and how they conceptually differ or overlap, we compared the collected definitions of key terms systematically with regard to the land use type(s) they declare either to expand or to reduce as a result of the phenomenon in question. Moreover, the definitions were screened for information suggesting a particular unit of measurement for quantifying the phenomenon to which they refer.

To assess the comparability of different land consumption and land take monitoring schemes, selected indicators were scrutinized with regard to their informative value and their methodological limits. To this end, underlying methodologies were screened and compared, with particular attention paid to the classification of land use types and land use changes. The compatibility of the selected international monitoring schemes was further checked by a comparison of land take trends that have been reported for the period 2009–2015 in the SDG context and for the period 2000–2018 in the EU context.

**3. Results**

In this section, we present a collection of published definitions of land consumption, land take, soil sealing, and artificialization, highlight similarities and differences between the concepts in focus, and examine the interrelations between the concepts land consumption and land take (Section 3.1). This is followed by information on relevant international monitoring and reporting systems and a comparison of selected conceptualizations and monitoring approaches at the national scale, highlighting differences between selected EU countries (Section 3.2).

*3.1. Definitions of Land Consumption, Land Take, and Related Concepts*

From the studied scientific and policy relevant literature, as well as some internet sources, we extracted 30 definitions of the concepts of interest. Among those were eight definitions of land consumption, 13 definitions of land take, five definitions of soil sealing, and four definitions of artificialization (or of derivations of these terms, respectively; Table 1). One of the definitions explicitly

equates land consumption and land take (#1; this and all subsequent numbers marked with # refer to Table 1). Another definition conceives land take as one of "three forms of land consumption" (#7).

Regarding the land use types they refer to, the extracted definitions range from broad to narrow (e.g., #1 vs. #2 for land consumption, or #16 vs. #9 for land take). Furthermore, the definitions vary in the degree to which they frame land take as being driven by urbanization or the growth of rural settlements (e.g., #2, #11), or by other specific developments like the building of infrastructures (e.g., #9 which was taken from a source dealing solely with the impacts of expanding transport networks). Many of the definitions of land consumption, land take, and soil sealing use the word "artificial" to specify the types of land, areas, or purposes under consideration (#5, #6, #8, #10, #12–#16, #20, #25, #26).

From the total set of collected definitions, one for land consumption (#1) encompasses the widest spectrum of land conversions and affected land types, including area used for (over-intensive) agriculture or forestry. In fact, according to this general definition of UN Habitat and the EEA, the term land consumption may be used to describe the spatial effects on land of any economic activity. In contrast, none of the collected land take definitions explicitly includes land conversions for the purpose of expanding agriculture or forestry. Rather, land take is described to convert (e.g., #10, #12) and diminish (#17) areas used for agriculture or forestry. Thus, if land consumption is conceived according to definition #1, all land take belongs to the category of land consumption, but not vice versa.

However, many of the reviewed sources conceive land consumption more narrowly than expressed in definition #1, i.e., as resulting primarily from urban developments (#2, #3, #5), and as referring only to processes that increase the area of "developed" or "artificial" land (#3–#6, #8). In these cases, the conceptual scopes of land consumption and land take are close, and possibly congruent; sometimes both concepts are used as synonyms (#11). In SDG indicator 11.3.1 (#3), the land consumption rate only refers to urban development [22].

Land take is described by several of the collected definitions as a process that converts, and thereby diminishes, natural, semi-natural, forest, or agricultural land (#10, #12, #17–#19). Additionally, two land take definitions explicitly include inner-rural development and densification (#11, #16). In contrast, the densification of existing settlements, including the redevelopment of brownfields, is excluded from the definition of land consumption and land take according to the SDG indicator 11.3.1 (#3) and the EU indicator of land take (#10).

With regard to soil sealing and artificialization, the collected definitions suggest that both concepts are less equivocal than land consumption and land take. All five definitions collected for soil sealing describe this phenomenon as involving the covering of land, and four out of five mention that impermeable/impervious materials are used for this purpose (#22, #24–#26). One of the definitions for soil sealing (#22) include processes by which soils are not necessarily covered but changed in such a way that they lose the ability to absorb water (e.g., due to compaction).

The four definitions collected for artificial or artificialized areas/land surfaces (#27–#30) characterize the land they refer to as being in use for a range of different human activities; the exact set of activities varies among the definitions. The most encompassing definition of artificialized land refers to all human activities other than agriculture and forestry (#29).

Beside the differences in land use types to which the definitions refer, variation was observed with respect to the units of measurements the definitions suggest for the phenomenon in question. Applying this criterion, the following types of definitions emerge: (i) definitions from which an appropriate unit of measurement cannot be inferred, e.g., because they attribute the term in question to a process or activity rather than to its result (e.g., #18, #22–#25); (ii) definitions suggesting that the phenomenon in question should be expressed in absolute terms, i.e., as the total amount of area affected (e.g., #12, #27–#30); (iii) definitions suggesting that the phenomenon in question should be expressed in relative terms, i.e., as a percentage (affected area compared to a reference area, e.g., #3, #5), a rate (change in affected area over time, e.g., #11, #20), or in relation to the human population (e.g., change in affected area per capita, e.g., #4, #6). Measures of land use change that refer to the human population have been termed land use intensity [12] or land use efficiency [29].

*3.2. Indicators for Monitoring Land Consumption and Land Take*

As mentioned in the introduction, the SDG indicator 11.3.1 is composed of the quotient of the land consumption rate (LCR) and the population growth rate (also referred to as the land use efficiency [16,46]). LCR is defined as "the percentage of current total urban land that was newly developed" [11] and calculated as (ibid.):

$$LCR = \frac{\ln\left(\frac{Urb_{t+n}}{Urb_t}\right)}{(y)}$$

where $Urb_t$ = total areal extent of the urban agglomeration in km$^2$ for past/initial year, $Urb_{t+n}$ = total areal extent of the urban agglomeration in km$^2$ for current year, and y = the number of years between the two measurement periods.

The methodology for computing the land consumption rate for the SDG indicator 11.3.1 is subject to a scientific debate [22]. Promising advances have been made in using new global datasets, tools, and maps, specifically those subsumed under the header "Global Human Settlement Layer, GHSL" [2,16,29,46]. However, there is no consensus yet on how to define the exact geographic boundaries of urban land and this poses a major challenge for assessing LCR [22,47].

The land take indicator has been defined by the European Environment Agency to address "the change in the area of agricultural, forest and other semi-natural land taken for urban and other artificial land development. Land take includes areas sealed by construction and urban infrastructure, as well as urban green areas, and sport and leisure facilities" [14] (see also definition #10 in Table 1). It captures all transitions of "agricultural areas," "forest areas," "wetlands," or "water bodies" into one of the following classes: "continuous urban fabric," "discontinuous urban fabric," "industrial or commercial units and public facilities," "road and rail networks and associated land," "port areas," "airports," "mineral extraction sites," "dump sites," "construction sites," or "sport and leisure facilities" [14].

According to this EEA methodology, land take may occur at any place where land classifies as natural, semi-natural, forest, or agricultural land, and these land type categories are almost exclusively assigned to land outside existing settlements. Urban green is seen as part of the settlement structure, and the respective land is regarded as "taken." It follows that developments within the boundaries of a settlement are not counted as land take, even if they affect vegetated plots (e.g., brownfields, gardens). Inner-urban areas affected by densification thus change from a less intensely developed state to more intensely developed state, but as the area has already been counted as "urban," the change has no effect on the land take statistics.

In contrast, successful land take compensation measures have an effect on the land take statistics because they change the status of areas from "taken" to "non-taken." Such a "land return to non-artificial land categories" [14] is also called re-cultivation, and can be achieved by unsealing or other conservation measures. By subtracting the re-cultivated area from the total area affected by land take, one arrives at the EEA indicator of *net land take* (see definitions #13 and #14 in Table 1). In the EU policy context, the concept of "net land take" plays a prominent role because it is the basis of the "no net land take until 2050" target. It is important to note that according to the EEA scheme, a distinction is made between "greening of urban areas" (which does not translate into a reduction of land take, because the area remains in the category "urban") vs. "re-naturalisation of artificial areas" (which is a compensation measure and does reduce land take) [14].

The EEA land take indicator is currently calculated using the CORINE Land Cover (CLC) Accounting Layers for the years 2000, 2006, 2012, and 2018. A serious limitation of CLC data is that small-scale developments cannot be detected in the Geographic Information System (GIS) vector dataset, as the Minimum Mapping Unit (MMU) for areal phenomena is 25 ha and the minimum width for linear phenomena is 100 m [48]. This leads to the exclusion of most linear transport infrastructures from the EU land take assessment [14] and may result not only in an underestimation of the total land take but also in flawed (counterfactual) judgements if, e.g., scattered sprawl (many

smaller developments across the landscape) scores much better than coordinated, large developments. Due to its low resolution, CORINE based measures have proven inadequate for detecting relevant developments on a small scale [49].

Furthermore, the CLC based method is not the only one by which land use is analysed across all EU Member States. The European Statistical Office (EUROSTAT), in close cooperation with the Directorate General for Agriculture and supported by the Joint Research Centre (JRC), samples the topsoil at more than 250,000 locations throughout the EU, approximately every three years. This survey is abbreviated LUCAS (which stands for Land Use/Cover Area frame statistical Survey) [50], and the obtained data are used, inter alia, for monitoring the progress towards the SDG targets, because this is under the responsibility of EUROSTAT.

The differences in the methodologies between CLC based (obtained from remote sensing) and LUCAS based estimates of land use changes (obtained from in-situ sampling) have resulted in divergent conclusions about recent trends in land consumption/land take: While the EEA detected a reduction of land take in the last decade (from over 1000 km$^2$/year between 2000–2006 to 539 km$^2$/year between 2012–2018) [14], EUROSTAT concluded that "the rate of land take has accelerated" between 2009–2015 [44].

In addition to the two international schemes described above, several different national classifications and monitoring systems exist to assess and monitor land consumption and land take. These have their own specificities and are not necessarily harmonized and aligned to the international reporting schemes. The national approaches listed in Table 2 demonstrate how the conceptualization of land take differs between EU countries. For example, gardens are judged as being neutral to land take in Czechia but regarded as part of the settlement area in Austria, Belgium (Flanders), and Germany, thus contributing to land take. Areas for which building plans exist but that are not (yet) actually used for purposes related to settlements, infrastructure, or service provision do not count as being taken in Belgium (Flanders), but contribute statistically to land take in Germany.

**Table 2.** Four approaches for assessing land take on the national scale: examples from of Austria, Belgium (Flanders), Czech Republic, and Germany [a].

| Country | Land-Take Equivalent in National Language | English Translation (by the Authors) | Definition | Further Specifications |
|---|---|---|---|---|
| Austria | Bodenverbrauch und Flächen inanspruchnahme | soil consumption and land occupation/utilization | Both exchangeable terms mean " . . . the permanent loss of biologically productive soil to building purposes for settlement and transport activities, recreational or disposal uses, as well as areas for mining, power plants or other similar intensive uses." [b] | The exchangeable use of Bodenverbrauch and Flächeninan spruchnahme derives from changing terminologies in the past years—leading to a use of both terms, even within one authority (like the Austrian Environment Agency). A distinction is made to soil sealing ("Versiegelung"), which describes the actual coverage of soil through asphalt or other building material, making it impermeable for water. |
| Belgium/Flanders | ruimtebeslag | settlement area | Areas affected by ruimtebeslag are the "[ . . . ] part of the space in which the biophysical function is not the most important. In other words, the space that is taken up by human activity (i.e., the space we use for housing, industrial, and commercial purposes, transport infrastructure, and recreational purposes). Parks and gardens, ecoducts across infrastructures, and some shoulders and banks along (road) infrastructures are also part of the settlement area." [c] | "This [ruimtebeslag] includes all plots of land with buildings (for residential use as well as for industrial and commercial use and for services), all land associated with road infrastructure, and all land used mainly for recreation. [ . . . ]. The built-up area within the military domains is included, but the exercise areas are not, because these often perform a (semi)natural function. "Land take," as understood in the Flemish definition, refers to the surface actually occupied by the mentioned use-categories." [c] |
| Czechia | zábor půdy | land take, land occupation | "Change in the area and structure of individual categories of agricultural land. The share of built-up and other areas in the total area." This is one of the indicators of sustainable development within the Czech Republic [44]. It implicitly assumes effective forest protection which holds in reality (the area of forest land increases in time). | The Czech statistical approach focuses on the amount of land with sealed surfaces which comprises two statistical categories: (i) built-up areas and (ii) other areas (mainly artificial land including transport infrastructure, landfills, or mining). Therefore, data on built-up areas within urban land are combined with those of outside urban land. Gardens (also within the boundaries of a city) are classified as a type of agricultural land and therefore are not considered as land taken by development within Czech Act No. 334/1992 Coll. On the protection of agricultural land [45]. |
| Germany | Flächenneu inanspruchnahme (für Siedlungs- und Verkehrszwecke) | extra/new land utilization (for settlements and transport purposes) | Land take in Germany is understood as the conversion of agricultural, forest, and other semi-natural and natural land into land for settlement and traffic. Land for settlement and traffic includes building areas and urban infrastructure as well as urban green areas and sport and leisure facilities and cemeteries (but excluding excavation areas). | In Germany, an area is statistically classified as a "settlement and traffic area" if it has been designated as a buildable area by a binding municipal land use plan, regardless of whether the area is actually used for this purpose (legal dedication determines the statistical classification). Thus, land take, as understood in the German definition, refers to the surface potentially occupied by the mentioned use-categories, resulting in a situation where land take happens when planning allows the creation of buildings or infrastructure [46]. |

[a] The displayed information was either obtained from the SURFACE questionnaire study or extracted from the referenced sources. [b] https://www.umweltbundesamt.at/umweltthemen/boden/flaecheninanspruchnahme; [c] https://www.statistiekvlaanderen.be/en/settlement-area-0.

## 4. Discussion

Over the past decade, a growing number of studies have addressed the operationalization of the concepts of land consumption and land take and related analytical questions [16,19,20,22,39]. This increased interest has been spurred by the fact that both concepts are components of internationally agreed sustainability indicators and are thus of high political relevance. Included in the analyses were inter alia, the factors driving the development of previously undeveloped land [27,51,52], the socio-economic context and spatial pattern of such conversions [53], and their potential consequences, e.g., for food production [1,54,55] and climate change adaptation [56]. However, as we discuss below, several theoretical and practical questions relevant for monitoring and curtailing land consumption and land take have not yet been fully explored.

Major conceptual ambiguities arise from the fact that different definitions are in use for the terms land consumption and land take (see Section 3.1). This partly confounds the relationship of the concepts with each other, and with related concepts, respectively. Notably, the inclusion of land conversions resulting from the expansion of (intensive) agriculture and forestry or from urban densification measures, within the meaning of land consumption and land take is controversial, whereas the contrast of naturalness vs. artificiality seems to be a key element of conceptualizing and operationalizing both concepts (Table 1).

### 4.1. Sharpening the Conceptualization of Land Consumption and Land Take

In the context of urbanization and spatial planning, land consumption and land take are frequently used (sometimes interchangeably) to describe the expansion of settlements, industries, and infrastructures at the expense of agricultural land, forests, or (other) semi-natural or natural areas e.g., [51,57] (see also Table 1). Thus, land consumption and land take often have the same meaning. The same is true for artificialization, a term that is used as an equivalent in some EU Member States [21], Table 1. Equating land consumption and land take seems legitimate and adequate in situations when exact and comparable quantifications for the respective phenomena are not the matter of discussion and an overall conceptual understanding is sufficient. For example, alternative planning objectives can be discussed in this way at a general level, such as avoiding settlement enlargement vs. creating new residential or commercial areas in the periphery. However, a change in the thematic focus may imply that land consumption is conceived differently and more broadly (see Section 3.1). If, e.g., deforestation for agricultural purposes is discussed, land consumption could be an appropriate term, but not land take.

Figure 1 synthesizes our understanding of the scopes of the discussed concepts and indicators in relation to different land use/land cover categories. In our view, land consumption and land take (along with artificialization) are conceptually broadly equivalent in the context of urbanization and spatial planning see also, e.g., [19], even though the exact spectrum of land conversions that are considered to contribute to these phenomena may vary. The ultimate categorization of a specific case contributing to, being neutral to, or compensating for the phenomenon in question depends on the exact definitions and methodologies applied, and is subject to national specificities (Table 2).

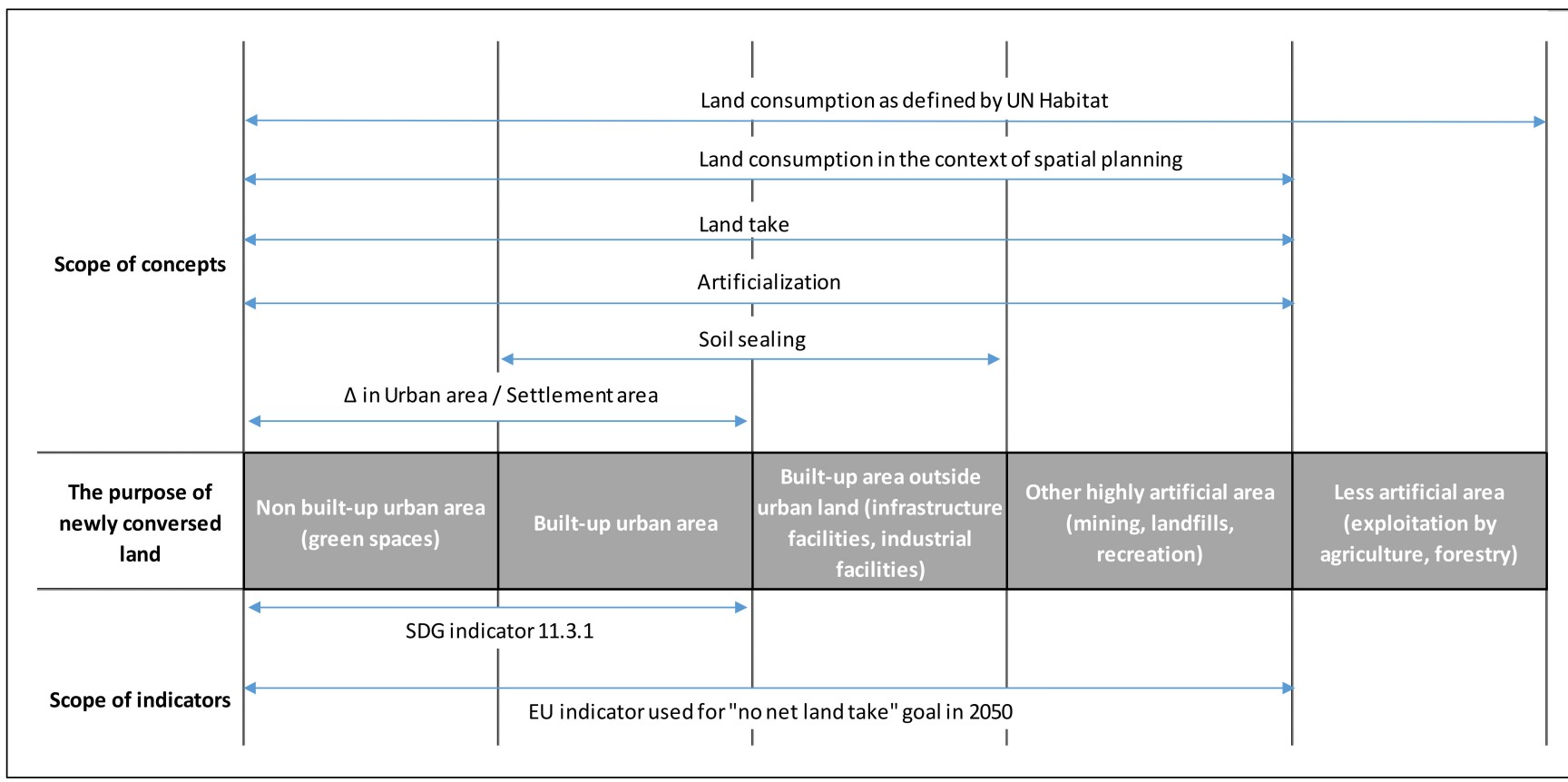

**Figure 1.** Scopes of the discussed concepts and indicators in relation to different land use/land cover categories.



However, some general statements can be made. Both concepts, land consumption and land take, emphasise that land conversions go along with the expansion or shrinkage of areas occupied by particular land use or cover types. The binary classification both concepts rest upon is coarse and largely neglects that two areas falling into the same broad use-category can perform very different ecological, economic, or social functions. In other words, land consumption and land take stress the quantitative rather than the qualitative aspects of land use changes. Both concepts are much more encompassing than soil sealing, which refers only to areas covered by impervious surfaces.

It is important to note that the developments that are of most interest here usually occur outside of (and thus between) existing settlements, often within the so-called peri-urban area and, in these cases, directly related to urban sprawl phenomena. We support the view that inner-urban development should not be conceived as contributing to land take and land consumption, as the affected areas are usually under strong anthropogenic influence and their soils compacted or otherwise significantly altered compared to their natural state. In fact, in the related policy discourse, densification (including brownfield redevelopment) is regarded as a measure for curtailing sprawl and urban expansion, and thus land consumption and land take [28,58,59].

One may argue that, compared to land consumption and land take, the term "artificialization" may be more intuitively understood and better suitable for conceiving land conversions as gradual processes. The latter would be compatible with the gradual conceptualisation of naturalness (as the counterpart of artificiality) that underlies the hemeroby concept about the degree of human impact on a natural environment [60,61]. However, determining the degree of artificialization or naturalness is inherently difficult, and certainly not reliably possible with remote sensing tools [19]. Furthermore, the discrimination of naturalness vs. artificiality carries a strong normative connotation. While this may also be true, to a lesser extent, for consumed vs. non-consumed, the distinction between taken vs. non-taken seems to be comparatively neutral.

*4.2. Monitoring Land Consumption and Land Take*

As the debatable case of urban densification shows, the evaluation of land use changes may depend on whether they occur inside or outside a particular area. The geographic and administrative localization of land use changes is furthermore critical for a clear attribution of responsibilities. Monitoring land use changes precisely is, therefore, a prerequisite for taking informed decisions and for post-hoc assessments of policy impacts. Respective indicators and monitoring tools need to be available and they are expected to deliver reliable and comparable data. As mentioned above, both concepts, land consumption and land take, are used in international reporting systems, but several methodological issues are still under debate [16,19,22,29]. In this context, it is critical to acknowledge that there is a great variety in land cover classification systems worldwide [62], and to specify which system has been applied in the case at hand. Furthermore, attention needs to be paid to the type of data that is used (information on land cover, land use, or administrative categories), and to the tools that are applied to obtain the data (remote sensing, field surveys, or cadastres). These factors considerably complicate the interpretation of respective indicators and potentially lower their comparability [19,36]. Thus, when the exact demarcation of an area that is consumed/taken is the subject of discussion, it is advisable not to use land consumption and land take synonymously, as this entails the risk of comparing two measures that have been estimated by diverging methodologies (see Section 3.2). In the context of SDG 11, land consumption has a specific meaning, as the land consumption rate is one of the components of SDG indicator 11.3.1. Here, "[ . . . ] the land type addressed [by the indicator] is urban land" [19], i.e., the indicator's scope is restricted to urbanization processes.

The discrepancy reported here between two examples of EU land take trends (assessed by EEA and EUROSTAT) further illustrated that a detailed methodological understanding of monitoring schemes is needed for interpreting quantitative measures and statements accurately as well as for disentangling related incongruences. This was further corroborated by our comparison of selected national approaches for land take monitoring (Table 2) which illustrated that, in the absence of

a uniformly shared definition, the understanding of land take varies between EU countries. Striking differences were observed with regard to the classification of gardens (that may or may not be part of the settlement area) and of areas that have been designated for development but not actually used (they may or may not contribute to land take). The Austrian/German example showed that conceptual differences between terms go beyond national language issues (both countries use almost the same wording) and are indeed a matter of national policy definitions.

Methodological sophistry also complicates the question of whether a conservation measure counts as a compensation for land take (and thereby reduces the net land take statistics, see Section 3.2). Compensation has become an essential instrument in spatial governance and is incorporated in the principles of sustainable land use for mitigating impacts: "avoid, reduce, compensate" [13]. While these principles are generally agreed upon, controversies arise as to whether damage caused to the living environment, specifically to ecosystems and biodiversity, can actually be counterbalanced. One may dispute which areas may count as equivalent to each other and whether a quality improvement of an area in one location can compensate for the loss of an ecosystem elsewhere. Moreover, when an area is taken, it is challenging to define what exactly needs to be compensated for (e.g., impacts to the scenery, the soils, biodiversity, or all of these?), and which relevant characteristics or components need to be considered (e.g., landscape elements, soil features, or species). Several other aspects are also far from certain, such as where compensation should take place (i.e., whether an area in close proximity should be prioritized over one farther away), how to deal with knowledge gaps regarding the damage caused, who decides about the equivalence of lost and gained components, or when to conclude that the loss is incommensurable and what consequences this has. These and related questions are at the centre of an intense debate around "biodiversity offsetting," which is discussed elsewhere [63,64]. Ultimately, these questions need to be negotiated and decided in a well-informed political arena or in a participatory planning setting; they are not primarily a matter of definitions and exact quantifications.

*4.3. Operating with the Land Take Concept—Potentials and Limits of Its Applicability*

For the sake of clarity and because our focus is on the EU policy context, we favour the term land take over the terms land consumption and artificialization in this subsection.

Land take is often associated with urbanization, but also the result of human activities and projects that may occur at great distances from cities and settlements. Examples include cross-country roads and industrial extraction or disposal sites (e.g., for the excavation of resources by open pit mining). These activities are all land based and, therefore, land take is usually considered to affect terrestrial systems applied as such, e.g., by [1]. However, according to the methodology for calculating the EU land take indicator, this includes wetlands, and the conversion of waterbodies to urban area is also counted as land take (see Section 3.2). While only a small fraction of the total land take affects wetlands and waterbodies in Europe (with exceptions in some countries, e.g., in Iceland or The Netherlands) [32], urbanization has a huge impact on waterbodies in some other parts of the world (examples include the expansion of some of the Asian megacities) [65]. The land take concept thus bears potential for wetlands as well. Moreover, and highly relevant to the EU context, infrastructures are increasingly built offshore. Examples include power lines, oil, and gas platforms and pipelines, mobile transmission towers, and communication cables. To assess their impact on the environment, it seems plausible to extend the applicability of the land take concept also to marine systems. It has already been used to calculate the spatial impact of offshore wind turbines [66].

Further complexity is added when environmental effects of above or below ground installations (such as storage tanks or solar panels) or on the surroundings of a project are considered. The latter has been referred to as indirect land take [67] and discussed, e.g., in the context of traffic routes causing the fragmentation of habitats [68]. We suggest that these are special cases to be decided upon according to the prevalent (national) classification system, and that non-binary categorizations might be most suitable for discussing them, as they have been introduced, e.g., for assessing the degree of hemeroby (as the opposite of naturalness) [60,61].

## 5. Conclusions and Recommendations

The simultaneous commitment of the EU to being "a frontrunner in implementing the 2030 Agenda and the SDGs" [69] and to striving for "no net land take" [18], calls for relating the two concepts *land consumption* and *land take* to each other and clarifying their scope of application. Based on our reflections above, we suggest working, on the one hand, toward a common general understanding of what land consumption and land take refer to (in the wider sense) and, on the other hand, toward a consistent interpretation and implementation of these concepts for monitoring and reporting purposes. This differentiation would facilitate the communication about land consumption or land take (given necessary specifications are provided) and, therefore, it should be popularised and constantly applied in practical, political and scientific contexts. Furthermore, it is equally important to use and clearly refer to existing definitions more rigorously and consistently, especially in the SDG- and EU context. Whenever accurate quantifications of trends are important, it is advisable not to use land consumption and land take synonymously and to conceive the respective phenomena as changes that occur over time, thus to consistently report them as a rate (e.g., as ha/day or $km^2$/year). Relating measurements to a reference figure (e.g., total area, population size) generally increases their comparability and should therefore be promoted. Urban densification measures should be excluded from the definition of land consumption (in the context of urbanization and spatial planning) and land take. In the EU context, we suggest prioritizing the term land take over land consumption, wherever appropriate. Exceptions may relate, e.g., to the international SDG framework.

Furthermore, the limitations of the concepts of land consumption and land take need to be duly considered—primarily their dichotomy, which neglects that there is a wide variation in effects of different uses on, e.g., ecosystem services. Alternative concepts, e.g., hemeroby, soil degradation, and possibly also artificialization, might prove more useful in this regard as they can be perceived as gradual processes.

Along with these primarily conceptual issues, it is also crucial to raise awareness of the scarcity of land in general, and of land with different qualities in particular. This heightened awareness should lead the broader public to question the processes by which land is assigned to different purposes and to reflect on the social and ecological impacts arising from these patterns.

In conclusion, arriving at a clearer conceptualisation of the key terms land consumption and land take would significantly facilitate the communication about the dynamics of settlement- and infrastructure-related land use within and across EU member states. It would allow a more in-depth exchange of experience gained in controlling urban expansion, soil degradation, and the loss of natural habitats, and a more efficient use of data. Moreover, a clearer conceptualization of the terms land consumption and land take is necessary for the consistent monitoring of policy goals, and thus for assessing and comparing the performance of the different country-specific, national-to-local, and inter-municipal spatial planning systems in place. These systems set different priorities and to some extent use different instruments [70]. Comparing their effectiveness with regard to reducing the adverse ecological, economic, and social effects of land take bears great potential for mutual learning from best-practice examples in a multitude of different political, geographic, and cultural contexts [71] and hence for more efficient spatial governance nationally and worldwide. All this requires reliable and comparable information on how and when land is developed for settlement- and infrastructure purposes, which in turn hinges on a harmonized application of key concepts.

Both indicators discussed above, the SDG indicator 11.3.1 and the EU indicator of land take, are most suitable for retrospective assessments of land conversions, and for post-hoc appraisals of policies and policy interventions. However, to continuously and prospectively ensure that a certain level of land take is not surpassed, as the "no net land take" target recommends, innovative and more targeted instruments are needed in spatial planning and governance. Examples of such instruments include trading schemes with planning permits [72] or with development rights adapted for the European context [73]. Spatial planning instruments also have the potential of allowing more nuanced judgements of land take. In spatial planning practice, the different economic and relational spatial

qualities of plots of land should be considered, since the development of pieces of land with the same dimensions can have very different qualitative consequences for sustainability. Accordingly, land take in the vicinity to a public transport line could, e.g., be given priority over land take along car-dominated transport lines or land take in poorly accessible peripheries. Additionally, the type of settlement in question should be taken into account; if land is taken, then spatial planning instruments should aim for high efficiency and high settlement density. Thus, complementing the "no net land take" target, spatial planning practice should try to achieve a qualitatively reasonable land take where needed [74]. Finally, coordinated and foresighted planning should be promoted as an effective tool for sustainable land management.

**Supplementary Materials:** The following are available online at http://www.mdpi.com/2071-1050/12/19/8269/s1, Table S1: The SURFACE expert questionnaire, Table S2: EU policy/science-policy documents considered.

**Author Contributions:** Conceptualization, E.M. and J.B.; methodology, E.M. and J.B.; formal analysis, E.M.; resources, J.G.i.F., A.H., E.J., N.M., L.P., B.R., R.R., J.S., M.T.S., and A.Y.; writing—original draft preparation, E.M., A.J., A.H., and J.B.; writing—review and editing, S.B., J.G.i.F., E.J., N.M., L.P., B.R., R.R., C.S.-S., J.S., M.T.S., E.V., and A.Y.; visualization, E.V.; funding acquisition, C.S.-S. and J.B. All authors have read and agreed to the published version of the manuscript.

**Funding:** This study was initiated at an international expert workshop hosted by the German Federal Environment Agency (UBA) as an element of the project "SURFACE—Standards and Strategies for the reduction of land consumption," also funded by the UBA; E.M. and J.B. received funding from this project (FKZ 3717181100).

**Acknowledgments:** S.B. acknowledges support by the German Federal Ministry of Education and Research (BMBF) in the framework of the funding measure "Soil as a Sustainable Resource for the Bioeconomy—BonaRes," project "BonaRes (Module B): BonaRes Centre for Soil Research, subproject A" (Grant 031A608A), and in the funding measure "Stadt-Land-Plus" through the project "Stadt-Land-Plus—Wissenschaftliches Querschnittsvorhaben" (FKZ 033L200). A.H. acknowledges funding by the Austrian Science Fund (FWF) through the project "Strategic Spatial Planning for Urban and Regional Shrinking" (FWF-J3993-G29). E.V. acknowledges funding by the Operational Program Research, Development and Education of the Czech Ministry of Education, Youth and Sports, supported by EU funds through the project "Smart City—Smart Region—Smart Community" (CZ.02.1.01/0.0/0.0/17_048/0007435). J.S. acknowledges funding of the Slovak Research & Development Agency (APVV-15-0136).

**Conflicts of Interest:** The authors declare no conflict of interest.

## Appendix A

**Table A1.** Countries covered by the SURFACE expert questionnaire survey.

| | Country | EU | Non-EU |
|---|---|---|---|
| 1. | Austria | ✓ | |
| 2. | Australia | | ✓ |
| 3. | Belgium (Flanders) | ✓ | |
| 4. | Brazil | | ✓ |
| 5. | Canada | | ✓ |
| 6. | China | | ✓ |
| 7. | Czech Republic | ✓ | |
| 8. | Denmark | ✓ | |

**Table A1.** *Cont.*

| | Country | EU | Non-EU |
|---|---|:---:|:---:|
| 9. | Estonia | ✓ | |
| 10. | France | ✓ | |
| 11. | Germany | ✓ | |
| 12. | Greece | ✓ | |
| 13. | Italia | ✓ | |
| 14. | Netherlands | ✓ | |
| 15. | Poland | ✓ | |
| 16. | Portugal | ✓ | |
| 17. | Romania | ✓ | |
| 18. | Slovak Republic | ✓ | |
| 19. | Slovenia | ✓ | |
| 20. | Spain | ✓ | |
| 21. | Switzerland | | ✓ |

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
