# Peer review of "Land Consumption and Land Take: Enhancing Conceptual Clarity for Evaluating Spatial Governance in the EU Context"

_sustainability, doi:10.3390/su12198269_

Round 1

Reviewer 1 Report

The research presented by authors has been devoted to an important issue - clarification of the terms “land consumption” and “land take” which is of high research, political and practical relevance. This research is important, comprehensive and new, it has high added value and will be interesting for scientists, PhD students and many other readers.

The article meets high requirements of the journal and is recommended for publishing.

Reviewer 2 Report

This is an exciting and important research exploring (in particular classifying and categorising) land consumption and the related concepts such as land take. The database is solid and extensive, with questionnaires and discussions from a focus group of international experts and a large number of literature and professional reports. The paper has the potential to become a highly-cited work in the field of environmental studies. This potential, however, has not been adequately achieved due to issues of the scope, structure, coherence and writing of the paper. My decision for the current manuscript is therefore a major revision with substantial changes. Please consider the following comments/suggestions positively; I am looking forward to reading a much-improved and -polished paper.

1. The paper starts with a workable target: reviewing and classifying the existing concepts related to land consumption; in other words, WHAT does the related field of research look like? However, the scope grow exceedingly large as the paper unfolds, in particular when arriving at the section 4 where five questions are raised largely to investigate HOW land take should be defined and measured. These are all important questions yet they seem to serve another paper. I would strongly suggest that the paper sticks to the workable target of reviewing and classifying existing concepts. Besides, the research data is mostly geared to the EU context instead of the international one (which could be a broader discussion). Therefore, the title and the related text could be changed to highlight the EU context.

2. The paper needs to be reconfigured with a clearer and simpler structure. First, the paper should identify a core concept, the ground based on which the comparison between different theories takes place. ‘Land consumption’ seems to be the core; ‘land take’, ‘soil-sealing’, ‘artificialisation’ and so forth (e.g. greenfield development, urban growth boundary) are related notions. Second, it is important to establish several criteria (e.g. content, focused space/scale, field of study, country of origin/use and measurement) when comparing these concepts related to land consumption. Table 1 is helpful in terms of showing what these concepts refer to; it is also confusing as different references are mostly piled up for each concept. A more analytical approach is called for, such as what Figure 1 shows.

3. The coherence of the paper could be enhanced. In several places in the text, readers are guided to jump from one subsection to another (e.g. the last sentence on page 5: ‘we discuss this delineation in detail in section 4.2’) – similar to ‘hyperlinks’. Instead, the revised manuscript should foster a more natural/smooth flow: using a coherent structure and writing to navigate the readers rather than creating such hyperlinks.

4. The writing of the paper needs to be improved. First, bullet points should be generally avoided in an academic paper. Second, many sentences are long and convoluted; try to replace them with shorter and clearer ones. Third, research methods could be introduced with more academic rigour/sense. For example, ‘focus group’ would be a better term than ‘international expert consultation’. ‘Literature review’ mostly refers to the review of academic writings (while this research also reviews professional reports); a better term for this study is ‘archival research’. Besides, details regarding why these two methods are selected and how they are conducted should be delivered clearer and simpler, with proper references to some methodology books. I am not going to point out the writing quirks and grammar issues considering the paper is expected to go through a structural change. I will hold such comments until the previous key issues are tackled.

Good luck with editing.

Reviewer 3 Report

This paper deals with a topic of considerable importance, develops a generally coherent analysis, and is generally well-written.

I recommend acceptance for publication. But I do have a few concerns that I would urge the authors to address in preparing a final text for publication:

  1. The introduction makes it sound like the paper will devote equal attention to the SDG 11 indicator of land consumption and the EU-wide land take indicator. But this is not really the case. The methods employed, the materials examined, and the analysis in the paper all focus far more on the EU case than on the SDG case. I'd recommend being forthright about this, making it clear that the primary focus is on the EU case with an effort to draw in the SDG case for the sake of comparison.
  2. I have some concerns about the methodology employed. The questionnaire produced only 25 responses, mainly from EU countries. The workshop involved only 20 experts, all apparently from EU countries, mostly overlapping with respondents to the questionnaire, and including most of the authors of the paper. This is okay as far as it goes. But it seems a somewhat shaky basis for drawing convincing inferences.
  3. The paper waffles on the relationship between land consumption and land take. It seems to suggest they are equivalent, but then goes on to indicate that this is not the case (pg 14). If I understand it correctly, all land take belongs to the category of land consumption but not vice versa. At a minimum, this needs to be acknowledged. It may also be appropriate to include an explicit discussion of the implications of this difference.
  4. I understand that the paper is largely an analysis of concepts, definitions, and indicators. That is fine as far as it goes. But it would help to talk more about the uses of these indicators. For example, is one indicator more useful than the other in answering certain types of questions? What sorts of opportunities does this exercise in conceptual clarification open up for substantive applications of interest from the perspective of policymakers?

There are a few minor glitches in the writing. But I assume these can be fixed easily in the process of preparing a final text for publication.

Round 2

Reviewer 2 Report

I am happy to read the revised manuscript, with clearer, more coherent, and more focused narratives and discussions. Also, most of the issues mentioned previously have been tackled. I therefore suggest that the paper is ready for publication once the following comments are considered and/or followed.

1. While the author clearly stated that ‘land consumption’ and ‘land take’ are both the focus of research, ‘land take’ seems to be prioritised. I think that a more accurate account of the research focus could be bridging the notions of ‘land consumption’ and ‘land take’ (this can be easily presented in the title, abstract, introduction, and results). Land consumption is more geared to SDGs at the global scale while land take is more suitable for the EU context – what could be considered an adaptive form SDG-11. This research seeks to show how these two key concepts are interrelated and how ‘land take’ diversifies in different EU countries.

2. In section 4.1, I would suggest that that the discussion on ‘artificialisation’ to be further reduced/removed or to be rephrased as a strand of the ‘land take’ concept. It is distracting and bizarre to show ‘artificialisation’ as the third significant concept in this subsection that seeks to (re)conceptualise ‘land consumption’ and ‘land take’.

3. It is necessary to continue to remove the writing quirks. For example, in line 204, ‘more than 50’ should be replaced by an exact number; in line 580, ‘conceptualising’ should be replaced by ‘reconceptualising’. Also, the remnants of the previous bullet-point structure should be further sifted out with a fine-tooth comb. For example, line 601 should be integrated with the following paragraphs, and try to avoid a sentence-as-paragraph (e.g. line 888 and 925).
